

# Mitochondrial genome characteristics of six *Phylloscopus* species and their phylogenetic implication

Chao Yang[1], Xiaomei Dong[2], Qingxiong Wang[1], Xiang Hou[1], Hao Yuan[3] and Xuejuan Li[2]

[1] Shaanxi Key Laboratory of Qinling Ecological Security, Shaanxi Institute of Zoology, Xi'an, China
[2] College of Life Sciences, Shaanxi Normal University, Xi'an, China
[3] School of Basic Medical Sciences, Xi'an Medical University, Xi'an, China

Corresponding author
Xuejuan Li, lixuejuan456@163.com

## ABSTRACT

The mitochondrial genomes of six *Phylloscopus* species—small insectivores belonging to the Phylloscopidae family—were obtained using the Illumina sequencing platform. The mitogenomes were closed circular molecules 16,922–17,007 bp in size, containing 13 protein-coding genes, 22 tRNA genes, two rRNA genes, and two control regions (*CR1* and remnant *CR2*). The gene orders were conserved in 35 sampled *Phylloscopus* mitogenomes in the GenBank database, with a gene rearrangement of *cytb-trnT-CR1-trnP-nad6-trnE-remnant CR2-trnF-rrnS*. The average base compositions of the six *Phylloscopus* mitogenomes were 29.43% A, 32.75% C, 14.68% G, and 23.10% T, with the A+T content slightly higher than that of G+C. ATG and TAA were the most frequent initiating and terminating codons, respectively. Several conserved boxes were identified in *CR1*, including C-string in domain I; F, E, D, and C boxes, as well as bird similarity and B boxes, in domain II; and CSB1 in domain III. Tandem repeats were observed in remnant *CR2* of the *Phylloscopus fuscatus* and *Phylloscopus proregulus* mitogenomes. A phylogenetic analysis with maximum likelihood (ML) and Bayesian inference (BI) methods, based on 13 protein-coding genes and two rRNA genes, indicated that the *Phylloscopus* species was divided into two larger clades, with a splitting time approximately 11.06 million years ago (mya). The taxa of *Phylloscopus coronatus*/*Phylloscopus burkii* and *Phylloscopus inornatus*/*P. proregulus* were located at the basal position of the different clades. The phylogenetic result of the *cox1* gene showed that *Seicercus* was nested within *Phylloscopus*. The complete set of mitogenomes of the *Phylloscopus* species provides potentially useful resources for the further exploration of the taxonomic status and phylogenetic history of Phylloscopidae.

## INTRODUCTION

Leaf warblers (*Phylloscopus*) belong to the family Phylloscopidae of the order Passeriformes (*Gill, Donsker & Rasmussen, 2022*). These small insectivorous songbirds have a wide distribution in the Old World (*Sun, Liu & Lu, 2020*). Leaf warblers have the highest number in Asia (*Alström et al., 2018*), with the greatest diversity in the eastern Himalayas and southern China (*Price, 2010*). There are currently 81 recognized species in the genus

*Phylloscopus* (*Gill, Donsker & Rasmussen, 2022*). Leaf warblers had been classified into two genera (*Phylloscopus* and *Seicercus*) (*Clement, Alström & Madge, 2006*); however, several phylogenetic studies have shown that traditional *Seicercus* is nested within *Phylloscopus*, and is also separated into two non-sister clades. Based on the phylogeny of mitochondrial and nuclear datasets, *Alström et al. (2018)* supported the synonymizing of *Seicercus* with *Phylloscopus*, leading to the synonymization of *Seicercus* and a monogeneric Phylloscopidae. The phylogenetic relationship and divergences of *Phylloscopus* have also been examined using multiple molecular markers such as mitochondrial genomes (mitogenomes) (*Yu et al., 2022*), mitochondrial genes combined with nuclear segments (*Alström et al., 2018*), and genome-wide data (*Zhang et al., 2021*). Other research has also been carried out on leaf warblers, in relation to topics such as their biogeographic history (*Johansson et al., 2007*), bioacoustic differentiation (*Calviño Cancela, Piña & Martín-Herrero, 2022*), and migratory behavior (*Adams et al., 2022*).

Several molecular markers, including mitogenomes, nuclear segments, ultra-conserved element (UCE) sequences, and genomic data, have been remodeled to study avian evolution (*Hosner, Braun & Kimball, 2016*; *Mackiewicz et al., 2019*; *Oliveros et al., 2019*; *Qu et al., 2021*; *Zhang et al., 2021*). Among them, mitochondrial DNA (mtDNA) has demonstrated maternal inheritance and haploidy, commonly containing 13 protein-coding genes (PCGs), two rRNA genes (rRNAs), 22 tRNA genes (tRNAs), and one non-coding region (control region (*CR*)) in the majority of birds. Mitogenome sequences have been widely used for evolution, phylogeny, population, and phylogeography studies of the genome characteristics of different bird orders such as Galliformes (*Li, Huang & Lei, 2015*), Charadriiformes (*Hu et al., 2017*), and Passeriformes (*Mackiewicz et al., 2019*), and have been effective markers for the exploration of evolutionary positions. By sampling ∼300 representatives of Passeriformes mitogenomes, *Mackiewicz et al. (2019)* analyzed four types of gene rearrangements, including a duplicated *CR* with adjacent genes, indicating that the duplication was the ancestral state and was maintained in early diverged lineages. Several complete mitogenomes of *Phylloscopus* are available in the GenBank database, such as those for *Phylloscopus tenellipes* (*Sun, Liu & Lu, 2020*; *Yu et al., 2022*) and *Phylloscopus proregulus* (*Jiao et al., 2018*), whose data have been used for the analysis of phylogenetic relationships of this genus and even the whole Passeriformes order.

In this study, the complete mitogenomes of six *Phylloscopus* species were sequenced using the Illumina sequencing platform. Based on these data, and by obtaining other *Phylloscopus* complete mitogenome sequence data from GenBank, we attempted to elucidate (a) the features and structures of the mitogenomes of six *Phylloscopus* species, and (b) the taxonomic status of the phylogenetic relationships and divergence times of the *Phylloscopus* species. The newly generated complete mitogenomes may be useful resources for further in-depth studies of the phylogenetic relationships of *Phylloscopus*, as well as Phylloscopidae.

## MATERIALS AND METHODS

### Sample collection, DNA extraction, and sequencing

Samples of six *Phylloscopus* species (*Phylloscopus burkii*, *Phylloscopus reguloides*, *Phylloscopus borealis*, *P. proregulus*, *Phylloscopus trochiloides*, and *Phylloscopus fuscatus*) were naturally deceased adults (Table S1, Fig. S1). Muscle tissue was collected, preserved in 100% ethanol, and stored at −20 °C in the Shaanxi Institute of Zoology, Xi'an, China. Genomic DNA was extracted using a DNeasy kit, and the library reconstruction followed the methodology of previous studies (*Yang et al., 2021*; *Yang et al., 2022*). The mitogenomes were sequenced using the Illumina HiSeq2000 and Xten platforms, with a paired-end read of 150 bp.

### Genome assembly and annotation

Mitogenome assembly was performed using MITOBim version 1.8 (*Hahn, Bachmann & Chevreux, 2013*) and MitoZ version 2.4 (*Meng et al., 2019*). Geneious version 11.1.3 was utilized for mitogenome sequence annotation (*Kearse et al., 2012*), with closely related species serving as references for comparison with the assembled results. Most tRNAs were identified using tRNAscan-SE version 1.21 (*Lowe & Eddy, 1997*), and the remaining tRNAs, rRNAs, and CRs were identified by comparison with other closely related species. The secondary structures of the tRNAs of *P. fuscatus* were referred to from the results of tRNAscan-SE version 1.21 (*Lowe & Eddy, 1997*) and other avian mitogenomes (*Yang et al., 2021*; *Yang et al., 2022*). The conserved elements in CR1 of *P. fuscatus* were analyzed by referencing previous studies (*Yang et al., 2022*).

### Characteristic analysis

The circular structures of the mitogenomes were plotted using the CGView Server (*Grant & Stothard, 2008*). The genome size and nucleotide composition were calculated using Geneious version 11.1.3 (*Kearse et al., 2012*), with the nucleotide bias calculated using the following formulas: AT-skew = (A −T)/(A+T) and GC-skew = (G −C)/(G+C) (*Perna & Kocher, 1995*). The relative synonymous codon usage (RSCU) and p-distance were evaluated using MEGA version 11 (*Tamura, Stecher & Kumar, 2021*). Tandem repeat sequences in the CRs were analyzed using the Tandem Repeats Finder version 20.10.2022 (*Benson, 1999*).

### Dataset construction, phylogeny, and divergence

On the basis of previous taxonomic relationships (*Mackiewicz et al., 2019*), 41 *Phylloscopus* individuals representing 19 species and one outgroup taxon (*Aegithalos concinnus concinnus*) mitogenome were downloaded from GenBank to construct the phylogenetic dataset (Table S2). The phylogenetic topologies were reconstructed using the two methods of maximum likelihood (ML) and Bayesian inference (BI) based on the combined 13 PCGs and 2 RNAs, with 14,005 bp after the alignment.

Each PCG was first aligned with amino acids using MUSCLE in MEGA version 11 (*Tamura, Stecher & Kumar, 2021*) and then transferred into nucleotide sequences. Two RNAs were also aligned using MUSCLE. The concatenated datasets were generated using SequenceMatrix version 1.7.8 (*Vaidya, Lohman & Meier, 2011*). The best-fit model

(GTR+I+G) was used to reconstruct the phylogenetic trees. The ML tree was created using IQ-TREE version 2.2.0 (*Nguyen et al., 2015*), with 5000 bootstrap replicates. The BI tree was analyzed using MrBayes version 3.1.2 (*Ronquist & Huelsenbeck, 2003*), with parameter settings of 2 independent runs with four simultaneous Markov chains for 2,000,000 generations and sampling every 100 generations. The first 25% of the generations was discarded as burn-in. The effective sample size (ESS) values were estimated using Tracer version 1.5 (*Rambaut, Suchard & Drummond, 2004*) with an ESS >200.

To further explore the phylogenetic relationships of leaf warblers, the mitochondrial *cox1* gene with a larger sampling size of 127 individuals belonging to 37 *Phylloscopus* species was analyzed using *A. concinnus concinnus* as the outgroup (Table S2). DNA sequences were aligned using MUSCLE, and Gblocks was employed to extract the conserved sites, which were used to construct the phylogenetic tree. The ML tree was reconstructed using IQ-TREE version 2.2.0 (*Nguyen et al., 2015*), with 5,000 bootstrap replicates.

The divergence time was estimated based on the *cytb* gene of the sampled *Phylloscopus* species. The Bayesian procedure was implemented using BEAST version 1.10.4 (*Drummond & Rambaut, 2007*). The calibration points were selected from the two major clades of the *Phylloscopus* species 11.7 million years ago (mya) (*Alström et al., 2018*) and 1.97 mya for *Phylloscopus canariensis* and *Phylloscopus collybita* using the TimeTree website (http://www.timetree.org/). The parameters were defined as the GTR+I+G substitution model, uncorrelated relaxed clock, lognormal relaxed distribution, and Yule prior in the analyses. The results of the runs executing 10,000,000 generations were used, with the first 25% discarded as burn-in values.

## RESULTS

### Mitogenome structure and organization

The whole mitogenomes of six *Phylloscopus* species were sequenced (GenBank accession no. MG681101, OR030349–OR030353); three *Phylloscopus* species (*P. fuscatus*, *P. reguloides*, and *P. trochiloides*) were the first new records for these species in the GenBank database. The three *Phylloscopus* species (*P. burkii*, *P. borealis*, *P. proregulus*) have been available in the GenBank database, with corresponding accession numbers of KX977449 in *P. burkii*, NC_045526 in *P. borealis*, and NC_037189 in *P. proregulus*, respectively (Table S2). Several contents, such as the length, nucleotide composition, and initiation and termination codon, were slightly different between sampled three species and that of the GenBank database. For example, the length of *P. borealis* was 16,898 bp in the sampled species, while 16,881 bp in the database, with corresponding nucleotide composition of 28.8% A and 33.4% C in this study, while 28.9% A and 33.3% C in the NC_037189. The initiation and termination codon of *nad3* (ATG and TAA) in *P. burkii* were different with that of KX977449 (GTG and TAG).

The total length of the mitogenomes ranged from 16,922 bp (*P. fuscatus*) to 17,007 bp (*P. reguloides*), similar to those of the other 35 *Phylloscopus* mitogenomes in the GenBank database (16,875–16,979 bp; Table S2). The length variation in the mitogenomes was probably due to the variable length of the control region, which was consistent with

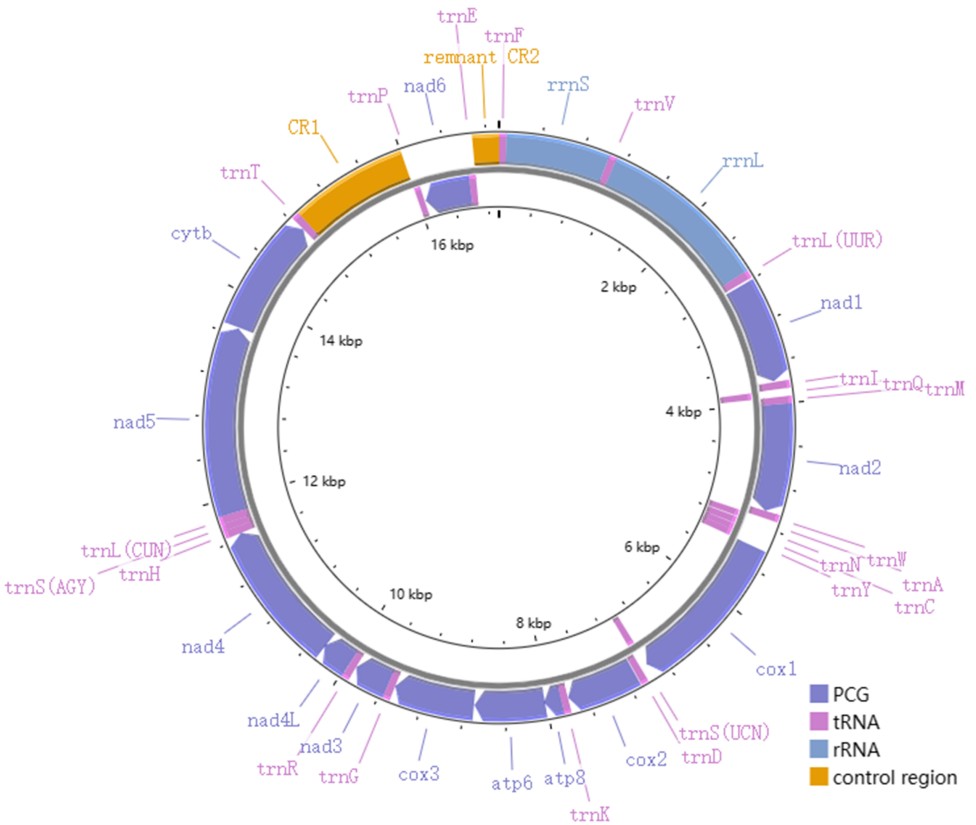

**Figure 1** The mitogenome organization of *Phylloscopus*.

previous avian mitogenomic studies (*Kundu et al., 2022*). The mitogenomes contained 37 genes, comprising 13 PCGs, 22 tRNAs, and two rRNAs, as well as two non-coding control regions (*CR1* and remnant *CR2*) (Fig. 1). Among them, nine genes (*nad6* and eight tRNAs) were identified on the N strand; the remaining 28 genes were identified on the J strand (Fig. 1). The gene order of the six *Phylloscopus* mitogenomes was identical, and similar to that identified in other birds, such as *P. proregulus* (*Jiao et al., 2018*) and *Alaudala cheleensis* (*Yang et al., 2021*). The base composition of the mitogenomes was C>A>T>G in the whole mitogenome (Fig. 2A), with average values of 29.43% A, 32.75% C, 14.68% G, and 23.10% T. The A+T content (ranging from 52.3% to 53.1%) was slightly higher than that of G+C, which is typical for avian mitogenomes.

## Protein-coding gene

The base composition of the PCGs (removing the termination codons) was C>A>T>G (Fig. 2B), with varied trends similar to those of whole mitogenomes. For different codon positions, the base composition C>A>G>T was observed in the first codons(Fig. S2A), T>C>A>G was observed in the second codons (Fig. S2B), and C>A>T>G was observed in the third codons (Fig. S2C).

All PCGs of the six *Phylloscopus* species were initiated with an ATG start codon (Fig. S3A). Eight PCGs (*nad1*, *cox2*, *atp8*, *atp6*, *nad3*, *nad4L*, *cytb*, and *nad6*) had TAA or TAG as

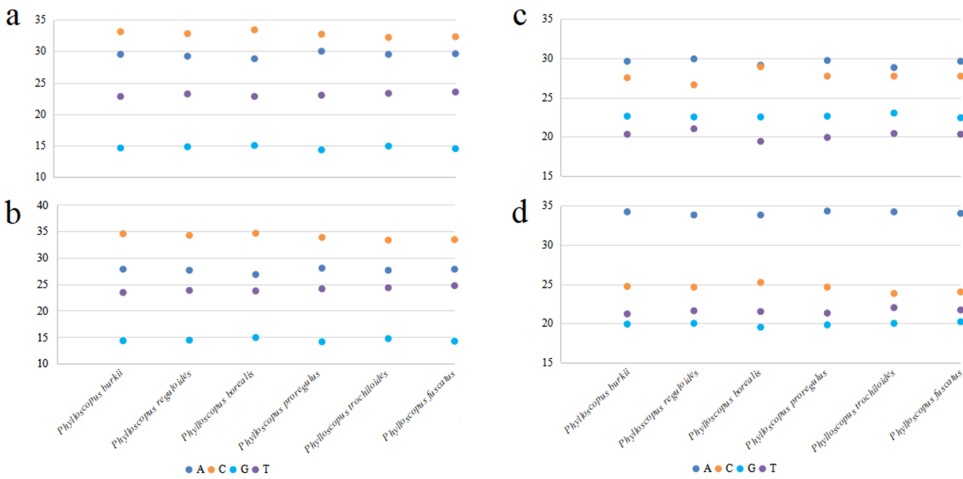

**Figure 2** **The base composition of mitogenomes in six *Phylloscopus* species.** (A) The whole mitogenome, (B) PCGs, (C) *rrnS*, (D) *rrnL*.

their termination codon, whereas AGG was the termination codon of *cox1* and *cox3*, AGA was the termination codon of *nad5*, GAA was the termination codon of *nad3*, incomplete TA was the termination codon of *nad2*, and T was the termination codon of *nad4* (Fig. S3B). The initiation codon of ATG and the termination codon of TAA are widely present in avian mitogenomic PCGs (*Morinha et al., 2016*; *Hu et al., 2017*; *Yang et al., 2022*). The incomplete T of *nad4* has also been observed in several avian PCGs such as *Pyrrhocorax pyrrhocorax* and *Pyrrhocorax graculus* (*Morinha et al., 2016*). The plausible explanation for incomplete codons is the post-transcriptional polyadenylation process, which adds 3′ A residues to the mRNA, generating a TAA stop codon (*Ojala, Montoya & Attardi, 1981*).

The RSCU values of the PCGs were similar in the six *Phylloscopus* species, with L2 having the highest value and L1 having the lowest value (Fig. S4). The whole RSCU content was similar to other avian mitogenomes (*Yang et al., 2021*). Codons ending with A and C were more frequent than those ending with U and G, as noted in previously reported research (*Yang et al., 2021*). CUA(L2), CGA(R), UCC(S2), and GCC(A) were the most frequently used codons (Fig. S4).

## RNA gene

All six *Phylloscopus* mitogenomes contained *rrnS* and *rrnL* of rRNA, and were located between *trnF* and *trnL* (UUR) and separated by *trnV* (Fig. 1). The total length of *rrnS* ranged from 974 bp (*P. fuscatus*) to 988 bp (*P. borealis*), and *rrnL* from 1,598 bp (*P. fuscatus*) to 1,602 bp (*P. burkii and P. borealis*). The base composition varied trend was A>C>G>T in *rrnS* (Fig. 2C), and the A+T content ranged from 48.5% (*P. borealis*) to 50.9% (*P. reguloides*). The base composition varied trend was A>C>T>G in *rrnL* (Fig. 2D), and the A+T content ranged from 56.2% (*P. trochiloides*) to 55.3% (*P. borealis*).

The length of the tRNA genes of the six *Phylloscopus* mitogenomes ranged from 66 bp (*trnS* (AGY)) to 75 bp (*trnL* (UUR)). In total, 21 tRNAs were folded into a classical clover-leaf secondary structure in the *P. fuscatus* mitogenome (Fig. S5). *trnS*

(AGY) lacked the dihydrouridine (DHU) arm, which is a common feature of avian mitogenomes in Passeriformes (*Gao, Huang & Lei, 2013*), Galliformes (*Li, Huang & Lei, 2015*), Charadriiformes (*Hu et al., 2017*), Piciformes (*Bi et al., 2019*), and Accipitriformes (*Jiang et al., 2019*). There were several pair mismatches in the stems; G-U was the most abundant type. Other patterns were observed, such as A-C in the amino acid accepter (AA) arm of trnF, C-C in the AA arm of *trnV*, and U-C in the anticodon (AC) arm of *trnG* (Fig. S5), which could be corrected by an RNA-editing process (*Lavrov, Brown & Boore, 2000*). When comparing all six *Phylloscopus* species, the loop sequences varied more often than those of the stems. Several stem sequences were completely conserved, such as the stems in *trnL* (UUR), *trnI*, *trnQ*, *trnM*, and *trnS* (UCN) (Fig. S6). The highest conserved site of the four arms of tRNA was the DHU arm, whereas the lowest was the AC arm (Fig. S6).

## Control region

The nucleotide composition of *CR1* was C>T>A>G (Fig. S7A); that of remnant *CR2* was C>A>T>G (Fig. S7B). The A+T contents of *CR1* and remnant *CR2* were slightly higher than in G+C, which is similar to that of *A. cheleensis* (*Yang et al., 2021*). The base skew results show that *CR1* had a slight T-skew and an obvious C-skew (Fig. S7C). Remnant *CR2* contained a medium A-skew and an obvious C-skew (Fig. S7D). The base composition and skew were similar to other Sylvioidea species (*Yang et al., 2021*).

Tandem repeats have previously been identified in avian mitogenomic *CRs* (*Ritchie & Lambert, 2000*; *Mundy & Helbig, 2004*; *Cho et al., 2009*; *Omote et al., 2013*; *Yang et al., 2021*; *Kundu et al., 2022*). Unique tandem repeat sequences were identified in both *Fregata minor* and *F. magnificens*, which could be used as species-specific markers (*Kundu et al., 2022*). For the tandem repeat sequences in the *CRs* of the six *Phylloscopus* mitogenomes, a consensus size of 46 bp (CATTTCATTAAACTCGCAAAGCCTACCAAACAACCGCATTCACACC) was observed in remnant *CR2* of *P. fuscatus*, with a copy number of 2.0. A consensus size of 45 bp (AACCAAACCTATCCCAAACCCCCCTCCCACTAAAAAACAAACAAA) was identified in remnant *CR2* of *P. proregulus*, with a copy number of 2.3.

*P. fuscatus CR1* could be divided into three domains, including ETAS (extended termination-associated sequence) domain I (nt 1–424), central conserved domain II (nt 425–859), and CSB (conserved sequence block) domain III (nt 860–1,104). This is similar to other avian mitogenomic *CRs* (*Bi et al., 2019*; *Yang et al., 2021*). Conserved box sequences have previously been identified in avian *CRs* (*Randi & Lucchini, 1998*; *Li, Huang & Lei, 2015*; *Aleix-Mata et al., 2019*). Several motifs have also been observed in *P. fuscatus CR1*, including C-string (CCCCCCCCCTCCCCCCCC) in domain I; F (GCGCTTCTCACGAGAACCGAGCTACTCAAT), E (GTTATTGGCGTCAGGGA CAT), D (CCTCCCGTGGTAACTTCAGGACCAT), C (CTGCCCTTCACTGATACTAGTGGTC GGTT), bird similarity (CACTGATGCACTTTG), and B (TCCCATTCATGGAC) boxes in domain II; and CSB1 (TATATAATGCAATGGTCACCGGACATG) in domain III. Sequences of the bird similarity box were completely conserved in *CR1* of the six *Phylloscopus* species. Motifs of CSB2, CSB3, the origin of heavy-strand replication (O$_H$), and light- and heavy-strand transcription promoters (LSP/HSP) were not identified in *P. fuscatus CR1*, which was similar to other birds (*Gao, Huang & Lei, 2013*).
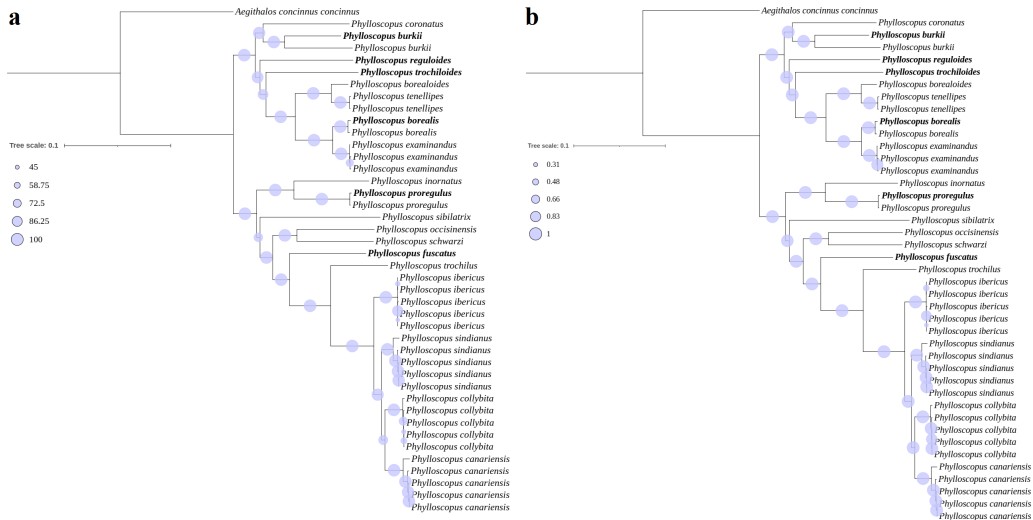

**Figure 3** **Phylogenetic trees reconstructed using PCG combining with RNA datasets.** (A) MI, (B) BI, the newly sequenced mitogenome sequences labelled with bold format.

## Phylogenetic and divergence analysis

The phylogenetic trees (ML and BI) revealed identical topologies that were supported by bootstrap values (BSs) in the ML tree (Fig. 3A) and posterior probabilities (PPs) in the BI tree at most nodes (Fig. 3B). The relationships of the genus *Phylloscopus* strongly supported a division into two major clades, including eight species contained in clade I (*Phylloscopus coronatus*, *P. burkii*, *P. reguloides*, *P. trochiloides*, *Phylloscopus borealoides*, *Phylloscopus tenellipes*, *P. borealis*, and *Phylloscopus examinandus*) and eleven species in clade II (*Phylloscopus inornatus*, *P. proregulus*, *Phylloscopus sibilatrix*, *Phylloscopus occisinensis*, *Phylloscopus schwarzi*, *P. fuscatus*, *Phylloscopus trochilus*, *Phylloscopus ibericus*, *Phylloscopus sindianus*, *Phylloscopus collybita*, and *Phylloscopus canariensis*) (Fig. 3). Compared with a previous study, the taxa included in clade I corresponded with clade α of the phylogenetic tree, based on one mitochondrial gene (*cytb*) and three nuclear segments (*ODC*, *myoglobin*, and *GAPDH*), whereas the taxa in clade II were consistent with clade β (*Alström et al., 2018*).

For clade I, *P. borealis* and *P. examinandus*, *P. borealoides* and *P. tenellipes*, and *P. coronatus* and *P. burkii* formed sister groups, with relatively higher BSs and PPs (Fig. 3). *P. burkii* traditionally belonged to the genus *Seicercus*; this species formed a strongly supported sister group with *P. coronatus* (BS = 98; PP = 1.0), which indicated that *P. burkii* was nested within *Phylloscopus*. The clade of *P. coronatus*/*P. burkii* was located at the basal position of clade I (BS = 100 and PP = 1.00). For clade II, *P. collybita* and *P. canariensis*, *P. occisinensis* and *P. schwarzi*, and *P. inornatus* and *P. proregulus* formed sister groups; these also contained higher BSs and PPs in the phylogenetic trees. The clade of *P. inornatus*/*P. proregulus* was at the basal position of clade II (BS = 100 and PP = 1.00).

A further phylogenetic tree based on the *cox1* gene also showed that two major cldaes contained in *Phylloscopus*, and the species traditionally classified within genus *Seicercus* was

nested within that of *Phylloscopus* (Fig. S8). The clade containing *Phylloscopus castaniceps* and *Phylloscopus grammiceps*, which used to belong to the *Seicercus* genus, was grouped with the *Phylloscopus* species (BS = 100).

The lowest p-distance value based on the PCG dataset was found between *P. borealis* and *P. examinandus* (0.0338), while the highest value was identified between *P. trochiloides* and *P. collybita* (0.1483) (Table S3). The relatively lower genetic distance between *P. borealis* and *P. examinandus*, and between *P. tenellipes* and *P. borealoides* (0.0353) showed closely phylogenetic relationship (Table S3), which consistent with the result of phylogenetic tree based on the PCG+RNA dataset with sister groups, respectively (Fig. 3). In addition, the lowest p-distance value based on *cox1* gene was identified between *Phylloscopus cebuensis* and *P. borealis* (0.00360), while the highest value was found between *P. ibericus* and *Phylloscopus* ijimae (0.16109) (Table S4). The relatively lower genetic distance between *P. cebuensis* and *P. borealis*, and between *Phylloscopus valentini* and *P. burkii* (0.01583) showed closely phylogenetic relationship (Table S4), which consistent with the result of phylogenetic tree based on *cox1* gene with sister groups, respectively (Fig. S8).

The divergence time results were similar to those in previous studies (Fig. S9) (*Alström et al., 2018*). The two clades split at 11.06 mya during the Tortonian stage of the Miocene, with a 95% HPD of 7.53–14.77 mya. This is similar to other results based on mitochondrial genes and nuclear segments (*Päckert et al., 2012*; *Alström et al., 2018*). The divergence times of the clade containing *P. canariensis*, *P. collybita*, *P. sindianus*, *P. ibericus*, and *P. trochilus* in clade II were analyzed. The divergence time between *P. canariensis* and *P. collybita* was 1.75 mya; between *P. sindianus* and *P. canariensis/P. collybit* it was 1.94 mya; between *P. ibericus* and *P. canariensis/P. collybit/P. sindianus* it was 2.29 mya; and between *P. trochilus* and *P. canariensis/P. collybita/P. sindianus/P. ibericus* it was 5.59 mya. These results are similar to those of *Alström et al. (2018)*.

The divergence times between the sister species ranged from 1.41 mya to 5.91 mya. In clade I, *P. borealis* and *P. examinandus* split at 1.64 mya, *P. borealoides* and *P. tenellipes* split at 1.41 mya, and *P. coronatus* and *P. reguloides* split at 5.91. In clade II, *P. collybita* and *P. canariensis* split at 1.75 mya, *P. occisinensis* and *P. schwarzi* split at 6.32 mya, and *P. inornatus* and *P. proregulus* split at 5.88 mya (Fig. S9). Among them, the sister groups between *P. borealis* and *P. examinandus*, *P. borealoides* and *P. tenellipes*, and *P. collybita* and *P. canariensis* represented relatively young *Phylloscopus* pairs, whereas those between *P. coronatus* and *P. reguloides*, between *P. occisinensis* and *P. schwarzi*, and between *P. inornatus* and *P. proregulus* were the old sister pairs.

## DISCUSSION

### Gene rearrangements

We observed that a gene rearrangement of *cytb-trnT-CR1-trnP-nad6-trnE-remnant CR2-trnF-rrnS* existed in the six *Phylloscopus* mitogenomes, which was different from the typical *cytb-trnT-trnP-nad6-trnE-CR-trnF-rrnS* of the avian gene order (*Urantówka et al., 2018*; *Mackiewicz et al., 2019*) (*e.g.*, *Gallus gallus*). Such gene rearrangements of the *Phylloscopus* species have also been observed in other birds (*Mackiewicz et al., 2019*; *Urantówka, Kroczak*

*& Mackiewicz, 2020*; *Yang et al., 2021*). Avian mitogenomic gene rearrangements, which include several gene orders, have been identified in previous studies (*Bensch & Härlid, 2000*; *Zhou et al., 2014*; *Eberhard & Wright, 2016*; *Caparroz et al., 2018*; *Kang et al., 2018*; *Urantówka, Kroczak & Mackiewicz, 2020*; *Yang et al., 2021*; *Kundu et al., 2022*), such as for Psittaciformes and Passeriformes (*Urantówka, Kroczak & Mackiewicz, 2020*). This could be explained by the tandem duplication–random loss (TDRL) model; the tandem duplication of *cytb-trnT-trnP-nad6-trnE-CR* was consistent with previous studies (*Yang et al., 2021*). Mitochondrial gene rearrangement provides useful information, such as elucidating the evolution of avian groups and their evolutionary relationships (*Kundu et al., 2022*).

### Two major clades and hierarchical nested structure

In clade I, the closer relationship of the sister group between *P. borealoides* and *P. tenellipes* (BS = 100; PP = 1.00) was consistent with previous studies based on mitochondrial genes and nuclear segments (*Johansson et al., 2007*; *Alström et al., 2018*; *Sun, Liu & Lu, 2020*; *Yu et al., 2022*). The sister group between *P. borealis* and *P. examinandus* (BS = 100; PP = 1.00) agreed with previous studies (*Alström et al., 2018*). Within clade II, a closer relationship between *P. inornatus* and *P. proregulus* (BS = 100; PP = 1.00) was observed, consistent with previous analyses utilizing concatenated mitochondrial gene datasets (*Jiao et al., 2018*; *Yu et al., 2022*). Furthermore, the genus *Seicercus* was nested within *Phylloscopus* (Fig. S8), which was consistent with previous studies (*Alström et al., 2018*).

### Divergence times

For Passeriformes birds, based on data for 4,060 nuclear loci and 137 families, *Oliveros et al. (2019)* observed that passerines originated on the Australian landmass (∼47 mya); the subsequent dispersal and diversification were affected by several climatological and geological events. For the *Phylloscopus* species, divergences have been analyzed in previous studies using a combination of mitochondrial genes and nuclear segments (*Moyle & Marks, 2006*; *Price, 2010*; *Alström et al., 2018*). According to previous studies, the diversification of leaf warblers took place ∼11–12 mya (*Johansson et al., 2007*). The main two clades ($\alpha$ and $\beta$) of *Phylloscopus* split approximately 11.7 mya (95% highest posterior density (HPD) = 9.8–13.7 mya) on the basis of the *cytb* gene. *Phylloscopus emeiensis*, *Phylloscopus neglectus*, and *Phylloscopus tytleri* are the oldest single-species lineages (7.3–8.3 mya) (*Alström et al., 2018*).

The divergence times between the sister species of *Phylloscopus* ranged from 0.5 mya to 6.1 mya, with the three youngest *Phylloscopus* pairs of sister species splitting at 0.5 mya, 0.8 mya, and 1.1 mya, respectively, and the three oldest strongly supported sister pairs splitting at 4.1 mya, 4.1 mya, and 6.1 mya, respectively (*Alström et al., 2018*). The differences may have been due to the sampling strategies and the lack of available whole mitogenome sequences of several *Phylloscopus* species, which resulted in a greater number of *Phylloscopus* mitogenomes being required to infer their separation events.

## CONCLUSIONS

We revisited the mitogenome features of six *Phylloscopus* species (*P. burkii*, *P. reguloides*, *P. borealis*, *P. proregulus*, *P. trochiloides*, and *P. fuscatus*), and investigated the evolutionary

characteristics of the gene order with a gene rearrangement of *cytb-trnT-CR1-trnP-nad6-trnE-remnant CR2-trnF-rrnS*. As well as the phylogeny in combination with other available *Phylloscopus* mitogenomes to analyze the evolution of *Phylloscopus*, the result indicated that the *Phylloscopus* species was divided into two larger clades, with a splitting time approximately 11.06 million years ago (mya). The phylogenetic result of the *cox1* gene showed that *Seicercus* was nested within *Phylloscopus*. Divergence times analyses investigated the differences of infer their separation events between young and old *Phylloscopus* sister pairs may have been due to the sampling strategies and the lack of available whole mitogenome sequences of several *Phylloscopus* species.

## ACKNOWLEDGEMENTS

We would like to special thanks to Fumin Lei of the Chinese Academy of Sciences for some sampling collection.

### Funding

This work was supported by the National Natural Science Foundation of China (Grant No. 31601846), the China Scholarship Council (202106875010), the Fundamental Research Funds for the Central Universities, China (Grant No. GK202304021), the China Postdoctoral Science Foundation (Grant No. 2016M602760), and the Western Young Scholars Projects of Chinese Academy of Sciences (Grant No. XAB2021YW01). The funders had no role in study design, data collection and analysis, decision to publish, or preparation of the manuscript.

### Grant Disclosures

The following grant information was disclosed by the authors:
National Natural Science Foundation of China: 31601846.
China Scholarship Council: 202106875010.
Fundamental Research Funds for the Central Universities, China: GK202304021.
China Postdoctoral Science Foundation: 2016M602760.
Western Young Scholars Projects of Chinese Academy of Sciences: XAB2021YW01.

### Competing Interests

The authors declare there are no competing interests.

### Author Contributions

- Chao Yang conceived and designed the experiments, analyzed the data, prepared figures and/or tables, authored or reviewed drafts of the article, and approved the final draft.
- Xiaomei Dong performed the experiments, analyzed the data, prepared figures and/or tables, and approved the final draft.
- Qingxiong Wang performed the experiments, prepared figures and/or tables, and approved the final draft.

- Xiang Hou performed the experiments, prepared figures and/or tables, and approved the final draft.
- Hao Yuan analyzed the data, authored or reviewed drafts of the article, and approved the final draft.
- Xuejuan Li conceived and designed the experiments, analyzed the data, authored or reviewed drafts of the article, and approved the final draft.

### Ethics

The following information was supplied relating to ethical approvals (i.e., approving body and any reference numbers):

We have adhered to all local, national, and international regulations and conventions, and we respect normal scientific ethical practices. The specimens used in this study collected from adult individuals that died naturally during field investigation, and have carried out harmless treatment. The animal experiment program of this project has been reviewed by animal Ethics Committee of Shaanxi Institute of Zoology (Xi'an, China), and conforms to animal protection, animal welfare and ethical principles, as well as relevant regulations of national ethical welfare of experimental animals.

### Data Availability

The mitogenome sequence data are available at NCBI GenBank: MG681101, OR030349–OR030353.

### New Species Registration

The following information was supplied regarding the registration of a newly described species:

The Phylloscopus species was divided into two larger clades, with a splitting time approximately 11.06 million years ago (mya). The taxa of P. coronatus/P. burkii and P. inornatus/P. proregulus were located at the basal position of the different clades. The phylogenetic result of the cox1 gene showed that Seicercus was nested within Phylloscopus.

### Supplemental Information

Supplemental information for this article can be found online at http://dx.doi.org/10.7717/peerj.16233#supplemental-information.

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
