# Peer review of "Mitochondrial genome characteristics of six Phylloscopus species and their phylogenetic implication"

_PeerJ, doi:10.7717/peerj.16233_

## Round 0.1 · original submission · Major Revisions

Your paper has been reviewed by three experts. While two of them liked your work one has recommended rejection. However, I would like to give you a chance to revise your paper based on the comments from all three reviewers.

Reviewer 1 ·

Basic reporting

The paper reports the sequence and annotation of six mitogenomes from the genus Phylloscopus and describe general features of these mitogenomes, such as gene order, nucleotide and amino acid compositions and codon usage. Additionally, the phylogeny of the genus Phylloscopus is reconstructed using the nucleotide sequences.
However, I found that many related species had been published in Genbank, but did not appear in this study. I suggest that the mitogenomes of all species of the genus Phylloscopus be analyzed together. So, Overall the paper needs to be rewritte. Although these research are interesting, they are rather limited and do not advance our knowledge of the subject sufficiently to warrant publication in Peerj.

Experimental design

no comment

Validity of the findings

no comment

Additional comments

no comment

Reviewer 2 ·

Basic reporting

This paper presents an analysis of the structural characteristics of the mitochondrial genome of six Phylloscopus species and elucidates the phylogenetic relationships within the Phylloscopus genus. The language employed in the paper is concise, the cited references are deemed appropriate and comprehensive, and the data provided is sufficiently detailed. Nonetheless, certain statements in the paper lack precision and would benefit from further refinement. It is recommended to revise and enhance these statements to ensure clarity and accuracy.

Experimental design

In this study, the Mitochondrial genome was sequenced utilizing the Illumina sequencing platform. The experimental design employed exhibited sound scientific reasoning, and the software was appropriately utilized for data analysis. The data obtained was comprehensively analyzed, ensuring a rigorous approach to the research.

Validity of the findings

This research outcome serves as a valuable supplement to the existing Mitochondrial genome data of Phylloscopus. It reveals the compositional characteristics of the Mitochondrial genome within the species and facilitates the classification of Phylloscopus into two distinct branches. These findings provide significant support, to a certain extent, to the previous research outcomes in the field.

Annotated reviews are not available for download in order to protect the identity of reviewers who chose to remain anonymous.

Reviewer 3 ·

Basic reporting

The article is good with clear objectives acceptable English language write-up.
The literature references are up to date and article structure are well organized.

Experimental design

Research questions are well defined and relevant. Although this is an extension of previous works been done by others, the approach is novel and highly informative.
Methodologies are well designed and appropriate.

Validity of the findings

The findings of this article is highly valid and should be important for future references on the taxonomy and systematic of the taxa.

Additional comments

Suggestion:
1) Provide pairwise genetic distance/divergence data between sequences/haplotypes in a table form
2) A photo of the six Phylloscopus species can be added as an additional figure.

---

## Round 0.2 · accepted · Accept

I have reviewed your revised manuscript. Thanks for addressing the reviewers' comments and suggestions. I think the manuscript is much improved.

Reviewer 2 ·

Basic reporting

no comment

Experimental design

no comment

Validity of the findings

no comment

Additional comments

Following the author's meticulous revisions, I am of the opinion that the manuscript has attained a level deserving of acceptance. The authors have diligently examined all the reviewers' comments and carried out comprehensive revisions in response. Hence, I recommend the acceptance of this paper.